



# Seasonal and Diurnal Dynamics of Subglacial Channels: Observations Beneath an Alpine Glacier

Ugo Nanni[1], Florent Gimbert[1], Christian Vincent[1], Dominik Gräff[2], Fabian Walter[2], Luc Piard[1], and Luc Moreau[3]

[1]University of Grenoble Alpes, CNRS, IRD, IGE, Grenoble, France
[2]Laboratory of Hydraulics, Hydrology and Glaciology (VAW), ETH Zürich, Zürich, Switzerland,
[3]Edytem, CNRS, Université de Savoie, Chambéry, France

**Correspondence:** Ugo Nanni (ugo.nanni@univ-grenoble-alpes.fr)

**Abstract.** Water flowing below glaciers exerts a major control on glacier basal sliding speeds. However, our knowledge on the physics of subglacial hydrology and its link with sliding is limited by lacking observations. Here we use a two-year long dataset made of on-ice measured seismic and in-situ measured glacier basal sliding speed records on the Glacier d'Argentière (French Alps) to investigate the physics of subglacial channels and its potential link with glacier basal sliding. Using dedicated theory

and concomitant measurements of water discharge, we quantify temporal changes in channels hydraulic radius and hydraulic pressure gradient. At seasonal timescales we observe, for the first time, that hydraulic radius and hydraulic pressure gradient present a four-fold increase from spring to summer, followed by a comparable decrease towards autumn. At low discharge during the early and late melt season channels respond to changes in discharge mainly through changes in hydraulic radius, a regime that is consistent with predictions of channels behaving at equilibrium. In contrast, at high discharge and high short-

term water-supply variability (summertime), channels undergo strong changes in hydraulic pressure gradient, a behavior that is consistent with channels being out-of-equilibrium. This out-of-equilibrium regime is further supported by observations at the diurnal scale, which demonstrate that channels pressurize in the morning and depressurize in the afternoon. During summer we also observe high and sustained basal sliding speeds, supporting that the widespread inefficient drainage system (cavities) is likely pressurized concomitantly with the channel-system. We propose that pressurized channels help sustain high pressure in

cavities (and therefore high glacier sliding speeds) through an efficient hydraulic connection between the two systems. Using the two regimes herein observed in channels seasonal-dynamics as constraints for subglacial hydrology/ice dynamics models may allow to strengthen our knowledge on the physics of subglacial processes.

## 1 Introduction

Subglacial water flow exerts a major control on glacier and ice sheet dynamics and their response to variations in water supply

(e.g. Iken and Truffe, 1997; Zwally et al., 2002; Sundal et al., 2011; Bartholomaus et al., 2011; Chandler et al., 2013; Hewitt, 2013; Joughin et al., 2018). Water flowing at the base of glaciers modulates glacier basal sliding speeds by lubricating the ice-bed interface. The higher the water pressure the weaker the basal friction, resulting in faster glacier sliding (Iken and Bindschadler, 1986; Schoof, 2005; Gagliardini et al., 2007). Water pressure does not simply depend on the total water input but



also on the way the water is conveyed through the subglacial drainage system (Lliboutry, 1968), a system that has, yet, yielded
limited observations (Flowers, 2015).

The subglacial drainage system of hard-bedded glaciers is considered to be two-fold. First, cavities form on the downstream
lee of bedrock bumps and are thought to enhance basal sliding through reducing the apparent bed roughness (Lliboutry, 1968).
These cavities constitute a widespread inefficient drainage system associated with high basal water pressure, slow water flow
(of the order of $10^{-2}$ m.s$^{-1}$, see e.g. Richards et al. (1996)) and limited hydraulic conductivity. Second, subglacial channels
form into the ice from conduit melt by flowing water heat dissipation, and close through ice creep (Röthlisberger, 1972; Nye,
1976). These channels constitute a localized efficient drainage system associated with lower basal water pressure, faster water
flow and higher hydraulic conductivity compared to within cavities. A drainage system for which a steady water input is routed
through channels tends to reduce basal sliding speeds compared to if water is predominantly routed through cavities (e.g. Foun-
tain and Walder, 1998; Schoof, 2010). Most of the current subglacial drainage models (Schoof, 2010; Hewitt, 2013; Werder
et al., 2013; Gagliardini and Werder, 2018) are based on this two-fold representation. These models succeed in capturing the
two-way channel-cavity coupling but still strongly rely on the choice of model parameters (e.g. cavities and channels hydraulic
conductivity, channels opening and closing rates, see de Fleurian et al., 2018). Observational constraints on these parameters
(e.g. water pressure, channel properties) and on the channel-cavity-sliding link are however very limited because of the limited
observations of the drainage system and concomitant measurements of basal sliding speeds (Flowers, 2015; de Fleurian et al.,
2018).

Direct observations of the drainage system have been relying on the analysis of water discharge measured near glacier outlets
(Collins, 1979; Hooke et al., 1985; Tranter et al., 1996, 1997; Anderson et al., 2003; Theakstone and Knudsen, 1989; Chandler
et al., 2013), of dye tracing experiments (Seaberg et al., 1988; Willis et al., 1990; Nienow et al., 1996, 1998), of recently
exposed subglacial environments (Vivian and Bocquet, 1973; Walder and Hallet, 1979) or of local water pressure boreholes
measurements (Hantz and Lliboutry, 1983; Copland et al., 1997; Sugiyama et al., 2011; Andrews et al., 2014; Hoffman et al.,
2016; Rada and Schoof, 2018; Gräff et al., 2019). These methods are mostly point-scale and often focus on the cavity-system
due to the very narrow extent of the channel-system (Rada and Schoof, 2018). As a consequence, quantitative information on
channels' long term temporal dynamics is limited, such that channels' properties (e.g. size, water flow velocity) and dynamics
(e.g. opening and closure rate) remain poorly constrained.

Interactions between channels and cavities are often observed indirectly from evaluating glacier flow-velocity variations
in response to meltwater supply variability. High and sustained water supply over long timescales (e.g. during the peak melt
season) has been observed to trigger glacier deceleration (Bartholomew et al., 2010; Sole et al., 2013; Tedstone et al., 2013,
2015). This behavior is related to the fact that channels-development increases the drainage system capacity and therefore re-
duces the average basal water pressure (Fountain and Walder, 1998). On the contrary, during short term water supply increase
(e.g. at the early melt season or at diurnal scales), glacier velocity changes have been observed to occur concomitantly with





water supply changes (Parizek and Alley, 2004; Palmer et al., 2011; Sole et al., 2013; Doyle et al., 2014; Vincent and Moreau,
2016). This behavior is mostly related to the pressurization of the cavity-system, causing average basal water pressure rise and
subsequent basal sliding speeds increase (e.g. Nienow et al., 2005; Schoof, 2010; Rada and Schoof, 2018). During periods of
well-developed channelized system (e.g. in summer), this behavior has also been observed because of a channelized system
drainage capacity being overwhelmed by the water input changes (Bartholomaus et al., 2008; Andrews et al., 2014) causing
pressurized channel flow. These studies have been capable to underline the overall differences between cavity and channel
control on subglacial water pressure over different timescales. However, the lack of robust channels observations independent
of those on cavities and concomitant with glacier sliding speeds measurements renders difficult a more quantitative characteri-
zation of the physics of subglacial hydrology and its link with sliding.

Here we use on-ice seismology to explore the evolution of subglacial channels over two complete melt seasons. Over the
last decade an increasing number of studies have shown the high potential of analyzing high-frequency (>1 Hz) ambient
seismic noise to investigate turbulent water flow and sediment transport in terrestrial rivers and streams (e.g. Burtin et al.,
2008, 2011; Tsai et al., 2012; Schmandt et al., 2013; Gimbert et al., 2014). The recent work of Gimbert et al. (2016) based on
observation of Bartholomaus et al. (2015) suggests that passive seismology may help filling the observational gap on the physics
of subglacial channels. Gimbert et al. (2016) adapted to subglacial channels a physical framework that describes how turbulent
water flow generates seismic waves and that was initially developed for rivers by (Gimbert et al., 2014). Contrary to rivers,
subglacial channels have the capability to be full and thus undergo pressurized situations. By applying this modified framework
to the Mendenhall glacier (Alaska) over a two-month long summer period, the authors show that one can use concomitant
seismic noise and water discharge measurements to continuously and separately quantify relative changes in channel hydraulic
pressure gradient and channel hydraulic radius. They observed that channels mainly evolve through changes in hydraulic
radius over long time scales (multi-weekly), whereas changes in hydraulic pressure gradient are often short-lived (sub-daily
to weekly). The use of such an approach to investigate channel physics on relevant glaciological timescales (e.g. diurnal and
seasonal) yet remains to be conducted and the resulting channels observations remain to be compared to other independent
observations, such as basal sliding speed, over such time scales. This is the objective of our study. To this end we conduct a
unique and almost uninterrupted two-years passive seismic survey on the Glacier d'Argentière (French Alps), together with
continuous measurements of subglacial water discharge, glacier basal sliding speeds and local subglacial water pressure. First,
we characterize the subglacial channel-flow-induced seismic power signature and use the model of Gimbert et al. (2016)
to derive timeseries of hydraulic pressure gradient and hydraulic radius. We then compare these channel properties to the
other independent measurements of glacier sliding speeds and basal water pressure. We also compare our seismically-derived
observations with the theory for subglacial channels physics proposed by Röthlisberger (1972) to assess the implications of
these observations for channels physics. Finally, we investigate the equilibrium state of subglacial channels to discuss the
channel-cavity interactions and their potential link with basal sliding throughout the melt season.





## 2 Rational

Here we provide a brief background on the theoretical framework of Gimbert et al. (2016), which relates seismic noise and water discharge to subglacial channel-flow properties, and that of Röthlisberger (1972), which predicts subglacial channel

hydraulic pressure gradient and hydraulic radius scaling as a function of water discharge under certain assumptions.

### 2.1 Theory of subglacial channel-flow-induced seismic noise

Turbulent water flow in a river or a subglacial channel generates frictional forces $F$ acting on the near boundaries (e.g. river bed or conduit wall), which in turn cause seismic waves with given amplitude and spectral signature (Gimbert et al., 2014). By propagating through a medium (e.g. rock, gravel or ice), seismic waves cause ground motion at any location $x$ away from the

source location $x_0$ (Fig. 1). The relationship between the force timeseries $F(t, x_0)$ applied at $x_0$ in a channel and the ground velocity timeseries $U(t, x)$ measured at $x$ can be described from Aki and Richards (2002) as

$$U(t,x) = F(t,x_0) \otimes \frac{dG(t,x;x_0)}{dt}, \tag{1}$$

where $G(t)$ is the displacement Green's function that converts the force applied at $x_0$ into ground displacement at $x$ and the notation $\otimes$ stands for the convolution operator. The seismic power $P$ of such signal is defined over a time period $T$ as

$$P(f,x) = \frac{U(f,x)^2}{T}. \tag{2}$$

where $U(f) = \mathcal{F}(U(t))$ is the Fourier transform of the ground velocity timeseries and $f$ is the frequency. We note $P_w$ the seismic power induced by turbulent water flow. Based on a description of the force $F(f)$ as a function of flow parameters, Gimbert et al. (2014) demonstrated that $P_w$ scales as

$$P_w(f) \propto \zeta(\frac{H}{k_s})W u_*^{14/3} \tag{3}$$

where $u_*$ is river bed shear velocity, $W$ is river width and $\zeta$ is a function that accounts for turbulence intensity changes with changes in the apparent roughness that depends on $H$ the flow depth and $k_s$ the wall roughness size (Fig. 1).

To relate $P_w$ to subglacial channels properties, Gimbert et al. (2016) expressed the shear velocity as $u_* = \sqrt{gRS}$ where $g$ is gravitational acceleration, $R$ the hydraulic radius and $S$ the hydraulic pressure gradient. The hydraulic radius $R$ is defined as

the ratio of the cross-sectional area of the channel flow to its wetted perimeter (Fig. 1). This parameter scales with flow depth for open channel-flow. The hydraulic pressure gradient $S$ defines the water pressure rate of change in the flow direction. This gradient exerts a strong control on water flow velocity: the greater $S$, the faster is the flow. For free surface flow $S$ equals channel slope. In a case of constant channel slope and channel geometry, increasing $S$ means closed and pressurizing channel-flow.





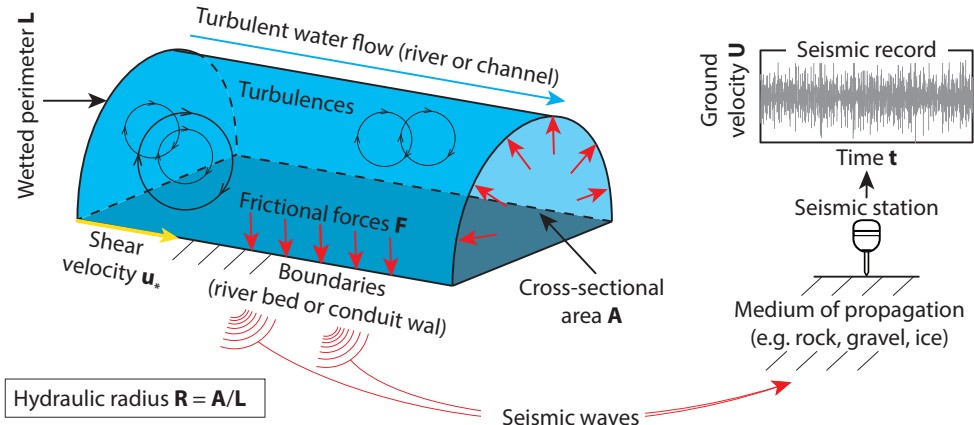

**Figure 1.** Schematic representation for the subglacial channel-flow-induced seismic noise. Representation of an idealized conduit of hydraulic radius $R$ with a bed shear velocity $u_*$ (see Eq.(3)). Turbulent flow generates frictional forces $F$ causing seismic waves and resulting in a ground velocity $U$ that is recorded at a seismic station (see Eq.(1)).

Gimbert et al. (2016) then expressed the water discharge $Q$ as a function of water flow velocity $V$ using the Manning-Strickler relation $U = \frac{R^{2/3}S^{1/2}}{n'}$ with $n'$ is the Manning's coefficient (Manning et al., 1890; Strickler, 1981). To study $P_w$ for a subglacial channel flow configuration, Gimbert et al. (2016) considered that the source-to-station distance is constant, such that changes in $P_w$ are not caused by changes in source (channel) position. Gimbert et al. (2016) then assumed a constant number $N$ of channels and thus neglected the dependency of $P_w$ on $N$. Here we include the dependency of $P_w$ on $N$ by considering that

all channels have equal hydraulic radius and hydraulic pressure gradient (i.e. are of similar size and position compared to the seismic station) such that

$$P_w \propto N\beta R^{14/3}S^{7/3} \tag{4}$$
$$Q \propto N\beta R^{8/3}S^{1/2}, \tag{5}$$

where $R$ and $S$ are average values over all channels and $\beta$ is a function of conduit shape and fullness that may be neglected

(see supporting materials of Gimbert et al. (2016) for details). Combining Eqs.(4) and (5) and neglecting changes in $\beta$ leads to the two following formulations for $P_w$,

$$P_w \propto R^{-82/9}Q^{14/3}N^{-11/3} \tag{6}$$
$$P_w \propto S^{41/24}Q^{5/4}N^{-1/4}. \tag{7}$$





From Eqs.(6) and (7) two end-member cases can be evaluated. If changes in discharge occur at constant channel geometry
(i.e. constant $R$ and $N$) from Eq.(6) we have

$$P_w \propto Q^{14/3}, \tag{8}$$

In contrast, if changes in discharge occur at constant hydraulic pressure gradient and channel number (regardless of whether the conduit is full or not) from Eq.(7) we have

$$P_w \propto Q^{5/4}. \tag{9}$$

Beyond these end-member scenarii, one can use measurements of $P_w$ and $Q$ to invert $R$ and $S$ using Eqs.(6) and (7) as:

$$S = S_{ref} \left( \frac{P_w}{P_{w,ref}} \right)^{24/41} \left( \frac{Q}{Q_{ref}} \right)^{-30/41} \left( \frac{N}{N_{ref}} \right)^{6/41}, \tag{10}$$

$$R = R_{ref} \left( \frac{P_w}{P_{w,ref}} \right)^{-9/82} \left( \frac{Q}{Q_{ref}} \right)^{21/41} \left( \frac{N}{N_{ref}} \right)^{-33/82}, \tag{11}$$

where the subset *ref* stands for a reference state defined over the considered period. In the following we consider $N$ constant to invert for $R$ and $S$, and later support that our inversions are not significantly biased by potential changes in $N$.

## 2.2 R-channels theory

To date, state-of-the art subglacial drainage models use the theories of Röthlisberger (1972) to describe subglacial channel dynamics (see de Fleurian et al. (2018) for model inter-comparisons). Channels described in these theories are assumed to be of semi-circular shape and to form into the ice through melt by heat dissipation from the flowing water and close through ice creep. A channel evolves at steady state with water discharge $Q$ if melt and creep rates change instantaneously with changes in
$Q$. A steady-state channel is at equilibrium with $Q$ if melt (opening) rate equals creep (closure) rate, in which case Röthlisberger (1972) predicts

$$R \propto Q^{9/22} \tag{12}$$

$$S \propto Q^{-2/11}. \tag{13}$$

For a steady-state channel not in equilibrium with $Q$ and that responds solely through changes in pressure gradient $S$ (i.e. $R$
is constant) Röthlisberger (1972)' equations show that:

$$S \propto Q^2. \tag{14}$$

Further details on these equations can be found in Supplementary Sect. S2. Later we compare our inversions of changes in $R$ and $S$ (using seismic observations) with changes in $R$ and $S$ as predicted by the theory of Röthlisberger (1972) for steady-state channels at equilibrium or not with water discharge.





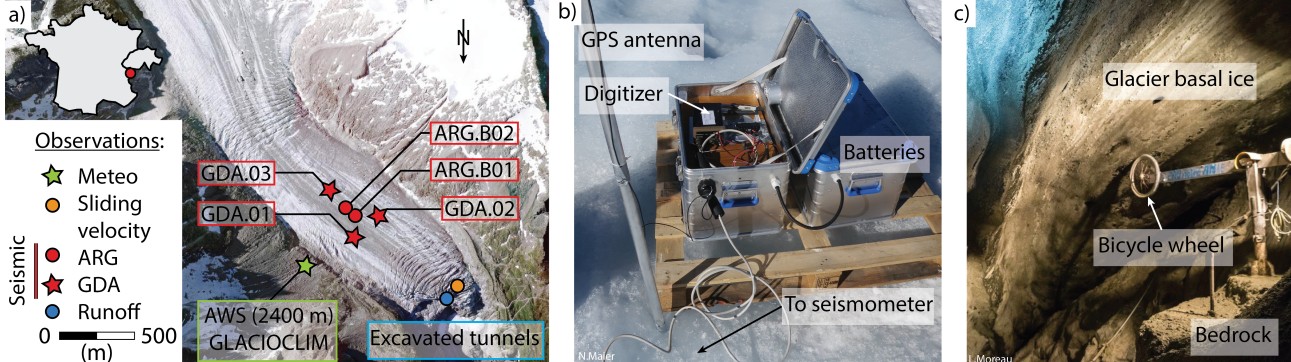

**Figure 2.** The Glacier d'Argentière monitoring setup (a) Aerial view of the Glacier d'Argentière field site (France) and location of the instruments used in this study. The aerial photography was taken in 2015. The seismic network is composed of the GDA (red circles) and ARG (red stars) borehole stations and is located according to positions in summer 2018. Station ARG.B02 is installed c. 70 m deep in the ice whereas the four other stations are installed c. 5 m deep. The GLACIOCLIM (https://glacioclim.osug.fr/) automatic weather station (green star, AWS) provides air temperature and precipitation. Basal sliding speeds (orange circle) and water discharge (blue circle) are measured thanks to direct access to the glacier base from excavated tunnels. Basal water pressure is measured at the same location as the basal sliding speeds. (b) Picture of the seismic instrumental setup used in this study. (c) Picture of the subglacial observatory with the bicycle wheel used to measure basal sliding speeds. [Photo credits: (a) IGN France, https://www.geoportail.gouv.fr/, (b) N. Maier, (c) L. Moreau].

## 3 Field setup

### 3.1 Site and glaciological context

The Glacier d'Argentière is a temperate glacier located in the Mont Blanc mountain range (French Alps, see Fig. 2). The glacier is 10 km long and covers an area of c. 12.8 km$^2$. It extends from an altitude of 1700 m above sea level (asl) up to c. 3600 m asl in the accumulation zone. Its cumulative mass balance has been continuously decreasing from -6 m water equivalent (w.e) in 1975 to -34 m w.e at present days with respect to the beginning of the twentieth century (Vincent et al., 2009). This site is ideal to study subglacial channels properties as it presents a typical U-shaped narrow valley (Hantz and Lliboutry, 1983) and hard bed conditions (Vivian and Bocquet, 1973), two conditions that favor a well-developed R-channel subglacial network (Röthlisberger, 1972).

In the present study we analyze the data recorded from spring 2017 to autumn 2018 with seismometers located between 2350 and 2400 m asl (Fig. 2). This location corresponds to the cross-section No. 4 monitored by the French glacier-monitoring program GLACIOCLIM (https://glacioclim.osug.fr/). There the glacier is up to c. 280 m thick (Hantz and Lliboutry, 1983, updated from a radar campaign conducted in 2018). Water discharge routing subglacially is monitored 600 m downstream of the seismometers at 2173 m asl near the glacier ice fall in subglacial excavated tunnels maintained by the hydroelectric power company Emosson S.A. Because of the valley-shape of the Glacier d'Argentière, subglacial water is almost entirely evacuated



through one major snout, as supported by observations of very limited water flowing elsewhere. Thus discharge measured at this location is well representative of the discharge subglacially routed under the seismometers location. Discharge measurements are conducted from mid-spring to early autumn with an accuracy of $0.01 \mathrm{m}^3\mathrm{s}^{-1}$ every 15 min by means of a Endress Hauser sensor measuring the water level in a conduit of known geometry. The minimum measurable value for water discharge is limited by the measurement accuracy and the maximum one is of $10 \mathrm{\ m}^3\mathrm{s}^{-1}$ due to the capacity of the collector. Because of sediments accumulation in the collector, flushes are recorded when the latter is saturated, causing spikes in the discharge record. We remove these spikes removing $Q$ values that present $\frac{d(Q)}{dt}$ higher than $0.2 \mathrm{\ m}^3$ per 15 min. Within the same tunnel network, a subglacial observatory is used to measure basal sliding speeds out of a bicycle wheel placed in contact with the basal ice (Vivian and Bocquet, 1973). Since August 2017 basal sliding speeds are measured at a time resolution of 5 s over a 0.07 mm' space segmentation. In the close vicinity a pressure sensor, of gauged type, is used to measure subglacial water pressure with 10 min time resolution and an accuracy of 0.004 Bar. The sensor is installed in a borehole drilled from the excavated tunnels towards the glacier bottom (see Vivian and Zumstein (1973) for details). Air temperature and precipitation measurements are obtained at a 0.5 h time step with the automatic weather station maintained by the French glacier-monitoring program GLACIOCLIM and located on the moraine next to the glacier at 2400 m asl. Precipitation is measured with an OTT Pluvio weighing rain gauge with a $400 \mathrm{\ cm}^2$ collecting area. When air temperature is below zero, only precipitation occurrences are accurate, but not total amount because of snow clogging.

## 3.2 Seismic instrumentation

We use five seismic stations installed in the lower part of the glacier (Fig. 2). The instruments belong to two seismic networks, denoted as GDA (3 stations) and ARG (2 stations). Stations GDA.01, GDA.02 and GDA.03 were deployed in Spring 2017 with c. 200 m inter-station distances. These stations have digitizers of the type Nanometrics Taurus, set to 16 Vpp sensitivity and a 500 Hz sampling rate, and sensors of borehole type (model Lennartz 3D/BH), with a lower Eigen frequency of 1 Hz. Station ARG.B01 was installed in October 2017 at the center of the GDA network at about 100 m from each GDA stations. The digitizer used for that station is a Geobit-SRi32L set to a 10 Vpp sensitivity and a 1000 Hz sampling rate. The sensor is of borehole type (model Geobit-C100) with a lower Eigen frequency of 0.1 Hz. Station ARG.B02 was installed in April 2018 about 50 m upglacier from station ARG.B01. The digitizer used for that station is a Geobit-SRi32 set to a 0.625 Vpp sensitivity and a 1000 Hz sampling rate. The sensor is of borehole type (model Geobit-S400), with a lower Eigen frequency of 1 Hz. All stations were installed c. 5 m deep below the ice surface, except ARG.B02 which was placed c. 70 m deep. Few data gaps occurred during our study due to difficulties in ensuring continuous power supply and data storage on glaciers.





## 4    Methodology

### 4.1    Calculation of seismic power at a 'virtual' station

The raw seismic record at each station is first corrected from its respective sensor and digitizer response. Then, the frequency-dependent seismic noise power $P$ is computed using the vertical component of ground motion (see Eq.(2)). $P$ is calculated with
the Welch's method over time windows of duration $dt$ with 50 % overlap (Welch, 1967). The longer $dt$, the more likely highly energetic impulsive events occur and overwhelm the background noise within that time window (Bartholomaus et al., 2015). To maximize sensitivity to the continuous, low amplitude, subglacial channel-flow-induced seismic noise and minimize that of short-lived but high energy impulsive events, we use a short time window of $dt$ = 2 s to calculate $P$, and average it over time windows of 15 min in the decimal logarithmic space. We express $P$ in decibel (dB, decimal logarithmic) which allows
properly evaluating its variations over several orders of magnitude.

We reconstruct a two-year long timeseries by merging records from the five available stations into one unique record at a 'virtual' station. To minimize site and instrumental effects on seismic power we shift the average power at each station to a reference one taken at ARG.B01. The seismic signal at our 'virtual' station is composed of the GDA seismic signals between
May 2017 end December 2017, and of the ARG seismic signals between December 2017 and December 2018 (see Fig. S1).

### 4.2    Evaluating bias due to anthropogenic noise

Later in section 5 we show that when water discharge $Q$ is low (in the early and late melt season) seismic power from anthropogenic noise ($P_A$) is comparable to the subglacial channel-flow-induced seismic power ($P_w$). Here we evaluate how much $P_A$ adding to $P_w$ can bias the evaluation of scaling predictions of Gimbert et al. (2016). We calculate the measured seismic power
$P_{mea}$ as $P_{mea} = P_A + P_w$ and $P_w$ as $P_w = Q^n$ with $n$ being equal to $\frac{5}{4}$ or $\frac{14}{3}$ as expected from theory (see Eqs.(8) and (9)). We quantify the relative contributions of $P_w$ and $P_A$ to $P_{mea}$ through the parameter $Sr$, which we define as $Sr = log\left[\left(\frac{Q}{P_{mea}}\right)^n\right]$. When $Sr$ tends to 1, subglacial channel-flow-induced seismic power dominates the measured seismic power and when $Sr$ tends to 0 anthropogenic noise power does.

In Fig. 3(a) we show $P_{mea}$ temporal evolution with a constant $P_A$ at 0 dB and a $P_w$ that responds to the evolving water supply $Q$. For $P_w \propto Q^{14/3}$ (Fig. 3(a), red and orange lines), $P_w$ dominates the contribution to $P_{mea}$ within c. 10 days from water supply start. For $P_w \propto Q^{5/4}$ (Fig. 3(a), black and green lines) $P_{mea}$ contains both $P_w$ and $P_A$ contributions during a period three times longer than for $P_w \propto Q^{14/3}$. The evolution of $Sr$ with respect to $P_{mea}$-$P_A$ (Fig. 3(b)) is the same for both the constant hydraulic pressure gradient (red line) and constant hydraulic radius (grey line) scenarii. For $P_{mea}$-$P_A$ > 2 dB, $Sr$ is
higher than 0.8, meaning that subglacial channel-flow-induced seismic power contributes by more than 80% to the measured seismic power. Later in Sect. 5.2 we define $P_A$ based on winter conditions when $P_w$ is negligible and use the condition $P_{mea}$-$P_A$ > 2 dB to define the periods where we investigate the subglacial hydraulic properties and calculate $P_w$ as $P_{mea}$-$P_A$.




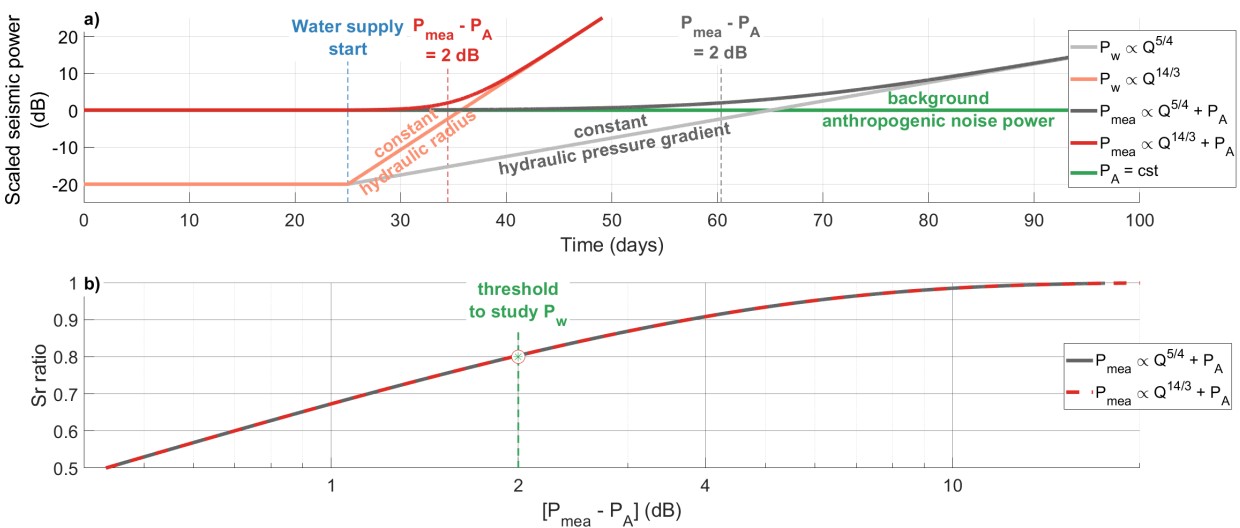

**Figure 3.** Bias evaluation of the anthropogenic noise. (a) Synthetic anthropogenic noise power (green line, $P_A$), synthetic subglacial channel-flow-induced seismic power $P_w$ for $n = \frac{5}{4}$ (grey line) and $n = \frac{14}{3}$ (orange line) and synthetic measured seismic power $P_{mea} = P_A + P_w$ for $n = \frac{5}{4}$ (black line) and $n = \frac{14}{3}$ (red line). (b) Evolution of $Sr$ ratio with respect to $[P_{mea}\text{-}P_A]$ for $n = \frac{5}{4}$ (grey line) and $n = \frac{14}{3}$ (red line). Note that the two curves overlap. See Sect. 4.2 for definition of $Sr$.

## 4.3 Definition of metrics to evaluate sub-diurnal dynamics

Since the $P_w$ versus $Q$ relationship is not unique and may vary with time (see Sect. 2), we expect that the diurnal timeseries of
$P_w$ versus $Q$ may exhibit different patterns throughout the melt season; and that these patterns reveal changes in the subglacial hydraulic properties. To systematically quantify the diurnal variability of $P_w$, $Q$, $R$ and $S$ throughout the melt season we define three appropriated metrics that we calculate on a daily basis (hydrological day). To focus on the diurnal variability only, we bandpass filter our timeseries within a [6-36] h range (see Appendix Fig. A1 for details). Our first metric quantifies the diurnal variability of a given variable $x$ during a day $d$ and corresponds to the coefficient of variation $C_v$ defined as:

$$C_v = \frac{\sigma(x_d)}{\overline{x_d}} \qquad (15)$$

with $x$ a given variable, $\sigma(x_d)$ its daily standard deviation and $\overline{x_d}$ its daily average. Our second metric $\phi$ quantifies diurnal hysteresis between $P_w$ and $Q$ by evaluating the difference between $P_w$ when $Q$ is rising, e.g. in the morning, and $P_w$ when $Q$ is falling, e.g. in the afternoon. Following the approach of Roth et al. (2016) we define $\phi$ as:

$$\phi = \frac{\overline{P_{w,d,rising}} - \overline{P_{w,d,falling}}}{\overline{P_{w,d,falling}}}. \qquad (16)$$

The larger $\phi$, the more seismic energy is recorded during the rising discharge period with respect to the falling one. Hysteresis can occur either because of an asymmetry between $P_{w,d,rising}$ and $P_{w,d,falling}$ or because of a time lag between $P_w$ and $Q$. To





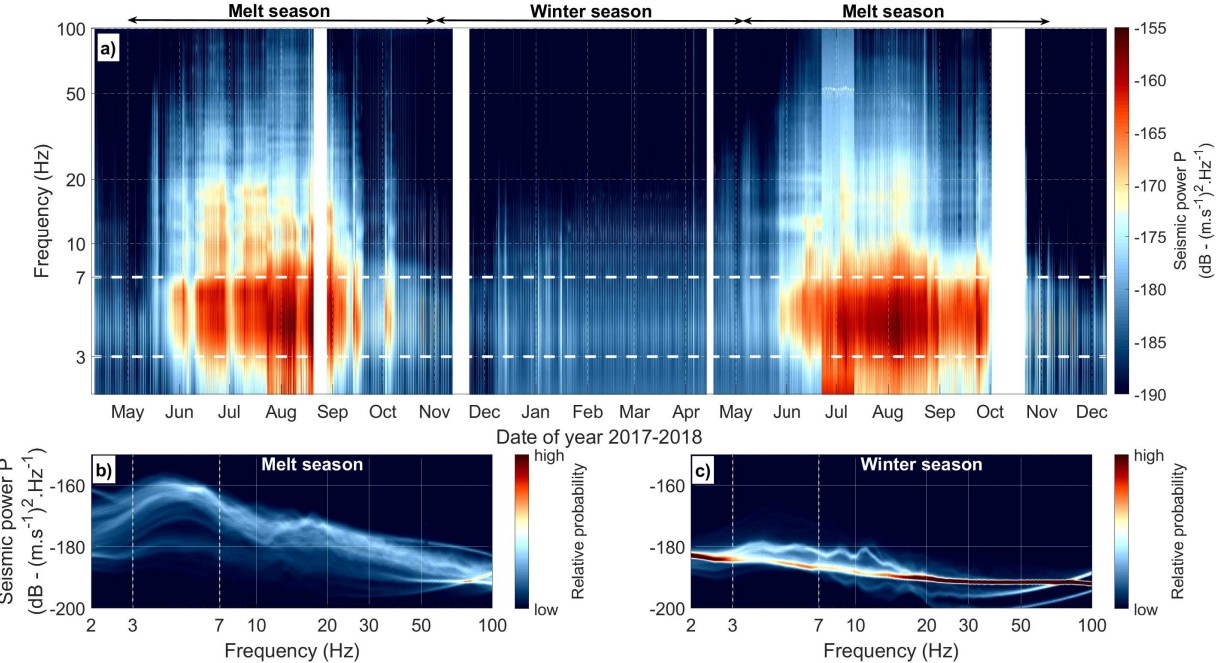

**Figure 4.** (a) Spectrogram of the seismic power *P* as a function of time (xaxis, May 2017 to December 2018) and frequency (yaxis, log-scale). White bands show data gaps. (b) and (c) Spectral distribution of seismic power during the melt seasons (b) and the winter seasons (c). Colors represent occurrence probability and are identical for (b) and (c).

avoid ambiguity between these two hysteresis sources our third metric corresponds to the daily time lag $\delta t$ between the time $t(P_{w,d,max})$ when $P_w$ is maximum and the time $t(Q_{d,max})$ when $Q$ is maximum and is defined as:

$$\delta t = t(Q_{d,max}) - t(P_{w,d,max}). \tag{17}$$

We set the condition that for $\delta t$ to be calculated, $t(P_{w,d,max})$ has to correspond to both the time when $P_w$ is maximum and has a null-derivative within a [-8, 8] h' time window around $t(Q_{d,max})$. We note that a time delay of about 0.04 h is expected due to water flowing at c. 1 m.s$^{-1}$ over the c. 600 m separating our seismic stations to where $Q$ is measured (see Fig. S2 for details). This means that any values of $\delta t$ greater than $\pm$ 0.04 h are not attributable only to water transfer time lags.

## 5  Results

**5.1  Overview of observations**

Seismic power *P* as calculated at our 'virtual' station based on records from our 5 stations (see Sect. 4) is shown in Fig. 4(a) as a function of time (May 2017 to December 2018) and frequency (2 to 100 Hz). Large seasonal changes in *P* are observed within

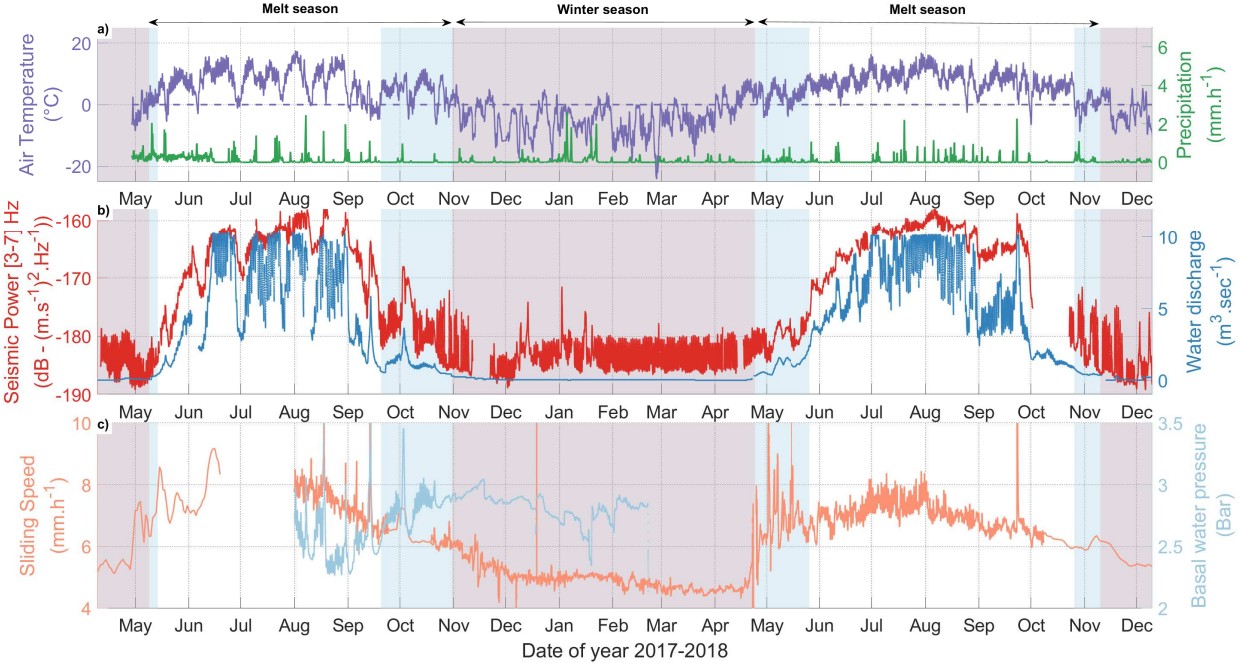

**Figure 5.** Timeseries of physical quantities measured from spring 2017 to winter 2018 at the Glacier d'Argentière. All data are averaged over 6 h. (a) Surface air temperature (purple line) and precipitation (green line) at the GLACIOCLIM AWS (Fig. 2). Dashed purple line shows $T = 0°C$ (b) Averaged seismic power within the [3-7] Hz frequency range from the 'virtual' seismic station (red line, $P_w$, see Sect. 5.2 for details) and subglacial water discharge (blue line, $Q$). (c) Basal sliding speeds (orange line) and subglacial water pressure (light blue line) measured at the Glacier d'Argentière subglacial observatory. Shaded red area represents the winter season, shaded blue area represents the period where the diurnal anthropogenic noise is too pronounced to study $P_w$ on a diurnal basis.

the [2-10] Hz frequency range, in which $P$ is higher by more than 2 orders of magnitude during the melt season (mid-May to September) compared to in winter. Changes in $P$ are also observed within the [10-20] Hz frequency range with $P$ during the melt season being about an order of magnitude larger than in winter. Significant changes of smaller amplitude are also observed at higher frequency ([20-100] Hz). Spectral distributions of $P$ presented in Figs. 4(b) and (c) show widely spread $P$ values during the melt season (Fig. 4(b), variations over more than 10 dB), as opposed to being comparatively much narrower in winter (Fig. 4(c), variations within 1-3 dB). Seismic power within the [3-7] Hz frequency range shows the highest range of variations from winter to summer (Figs. 4(a) and (b)). Over the two years, the overall spectral pattern remains similar, although intra-seasonal variations of $P$ during the 2017 melt season are more pronounced compared to the 2018 melt season.

The observed meteorological and hydrological conditions at Glacier d'Argentière together with the measured basal sliding speeds and the seismic power $P_{mea[3-7]\,Hz}$ as averaged within the [3-7] Hz frequency range are shown as a function of time



(May 2017 to December 2018) in Fig. 5. Water discharge $Q$ shows a strong seasonal signal with discharge lower than 0.1
$\mathrm{m^3.s^{-1}}$ in winter and up to values higher than 10 $\mathrm{m^3.s^{-1}}$ in summer. These changes are consistent with air temperature values,
and occur concomitantly with the evolution of $P_{mea[3-7]\,\mathrm{Hz}}$ (Fig. 5(b)). Further details on the comparison between $P_{mea[3-7]\,\mathrm{Hz}}$
and $Q$ are presented in Sect. 5.2. Over the first months of the melt season (early May to mid-June 2017 and late April to
mid-June 2018) $Q$ increases by about 2 orders of magnitude from 0.1 to to 10 $\mathrm{m^3.s^{-1}}$. At the same time, the amplitude of the
diurnal variations in $Q$ increases up to 3 $\mathrm{m^3.s^{-1}}$ over the summer. The evolution of basal sliding speeds presented in Fig. 5(c)
depicts a rapid acceleration from 5 $\mathrm{mm.h^{-1}}$ in May 2017 and April 2018 to 7 $\mathrm{mm.h^{-1}}$ over the following month. Sliding
speeds then stay almost constant through the summer, and slowly decrease down to February when they reach a minimum
value of 4.5 $\mathrm{mm.h^{-1}}$ (see also comparable observations made by Vincent and Moreau (2016) for the past decade). Basal water
pressure measurements (Fig. 5(c)) show that at the seasonal timescale the basal water pressure tends to be higher in winter
than in summer by c. 0.25 $\mathrm{Bar}$. In summer (August to mid-October 2017) the short-term (diurnal) variability in the basal water
pressure is more marked than in winter, as observed for the water discharge (Fig. 5(b) and Fig. A1). During heavy rainfall
(Fig. 5(a)) and consequent discharge (Fig. 5(b)), the basal water pressure and the sliding speeds are well in phase (Fig. 5(c)).
This evolution of the measured basal water pressure rather depicts a local behavior whereas changes in the basal sliding speeds
(Fig. 5(c)) rather represent average changes in the average basal water pressure conditions over our study area and therefore
better represent the global cavity-system pressure conditions.


    Measurement artefacts are observed for $Q$ with values being thresholded at 10 $\mathrm{m^3.s^{-1}}$, and for $P$ in July 2018 with high
seismic power values observed over the whole frequency range that we associate with the initially weak ice-sensor coupling of
ARG.B02. Site specificity of the GDA network used in 2017 causes higher seismic power in the [8-20] Hz frequency band in
2017 than in year 2018. These artefacts appear to do not significantly affect neither $P$ within the [2-10] Hz frequency range,
nor the concomitant temporal evolution of $P$ and $Q$ over the two years.

## 5.2    Seismic power induced by subglacial channel-flow

We consider seismic power $P_{mea[3-7]\,\mathrm{Hz}}$ averaged within the [3-7] Hz frequency range (Fig. 5(b) (red line)) as best repre-
sentative of subglacial channel-flow-induced seismic power $P_w$ because it shows the highest range of variations in response
to changes in $Q$ (Figs. 4 and 5). A similar frequency-signature of the subglacial channel-flow-induced seismic noise as been
observed by Bartholomaus et al. (2015), Preiswerk and Walter (2018) and Lindner et al. (2019) in glacial environments. This
frequency range is also comparable to those observed for terrestrial rivers (Burtin et al., 2008; Schmandt et al., 2013). As
$Q$ increases from less than 0.1 $\mathrm{m^3.s^{-1}}$ in early May to about 10 $\mathrm{m^3.s^{-1}}$ end of July, $P_w$ increases by up to 30 $\mathrm{dB}$ (i.e. 3
orders of magnitude). The relative inter-station variations of $P_w$ are lower than 0.5 $\mathrm{dB}$ even during periods of high discharge
(Fig. S2). This supports the accuracy and validity of our 'virtual' station reconstruction to study the subglacial channel-flow-
induced seismic power (Sect. 4). Variations in $P_w$ follow those of $Q$ during the melt season and over seasonal to weekly times
scales (Fig. 5(b)). Both the high sub-monthly variability in $Q$ and temperature observed in 2017 and the rapid changes in $Q$
occurring in fall 2017 and 2018 are also observed in the temporal evolution of $P_w$. In winter we observe high seismic power





bursts from December to mid-January occurring when $Q$ is null but concomitantly with the beginning of heavy snowfall events. These bursts are not associated with subglacial channel-flow-induced seismic noise but likely correspond to repeating stick-slip

events triggered by snow loading similar to those observed previously by Allstadt and Malone (2014). When $Q$ is lower than 2 $\mathrm{m}^3.\mathrm{s}^{-1}$ during winter, early spring and fall, we observe the superposition of regular weekly and daily variations in $P_{mea[3-7]\,\mathrm{Hz}}$ (Fig. 5(b)). This regular pattern corresponds to anthropogenic noise, as previously observed by Preiswerk and Walter (2018) in a similar setup.

Based on the condition proposed in Sect. 4.2 ($P_{mea}$-$P_A$ > 2 dB) we use the periods [May $10^{th}$ - November $1^{st}$] 2017 and [April $25^{th}$ - November $11^{th}$] 2018 to investigate the subglacial hydraulic properties (white and blue areas in Figs. 5 and 8). During these periods we subtract the mean winter diurnal pattern of $P_A$ (defined between January $29^{th}$ and April $4^{th}$ 2018) from $P_{mea[3-7]\,\mathrm{Hz}}$ to obtain $P_w$ (Fig. S3). At the diurnal scale, because $P_A$ can vary from day to day (week day, week end, holidays), the periods of very early and late melt season are still strongly influenced by $P_A$. To study diurnal changes in $P_w$

without being biased by anthropogenic noise we limit our analysis to the periods [May $15^{th}$ - September $22^{st}$] 2017 and [May $27^{th}$ - October $28^{th}$] 2018 (based on direct observation shown in Fig. S3; white areas in Figs. 5 and 8).

### 5.3 Comparison of observations with predictions from Gimbert et al. (2016)

#### 5.3.1 Analysis of seasonal changes

Seasonal scale observations and predictions of the subglacial channel-flow-induced seismic power $P_w$ versus water discharge $Q$ are shown in Fig. 6. We find that theoretical predictions from Gimbert et al. (2016) (red and black lines) are consistent with our observations (colored dots), which exhibit a general trend between that predicted at constant hydraulic pressure gradient (Fig. 6, see black lines calculated using Eq.(7)) and that predicted at constant hydraulic radius (Fig. 6, red lines calculated using Eq.(6)). As $Q$ increases at the very onset of the melt season (in end of April), observed $P_w$ values follow the trend of constant

hydraulic pressure gradient (Fig. 6 ①). As $Q$ increases more rapidly from mid May to end of June (Fig. 5(b)), $P_w$ follows a different trend of evolving hydraulic pressure gradient (Fig. 6 ②). The general trend from July to September is then dominated by changes in hydraulic radius (Fig. 6 ③). As $Q$ decreases during the melt season termination, observed $P_w$ values follow the trend of evolving hydraulic pressure gradient in a similar manner as during the early melt season (Fig. 6 ④). At the end of the melt season 2018 (Late October to November) our observations also show a trend of changing hydraulic radius although this

observation is not as clear in 2017 (Fig. 6 ⑤). A clear counter-clockwise seasonal hysteresis of up to 10 dB power difference is observed in Fig. 6 between $P_w$ and $Q$. This shows that for a similar water discharge, higher subglacial channel-flow-induced seismic power is generated in the late melt season compared to in the earlier melt season. The 10 $\mathrm{m}^3\mathrm{s}^{-1}$ threshold in $Q$ is well observable for the two years but does not bias the observed scaling of changing hydraulic radius observed during summer.





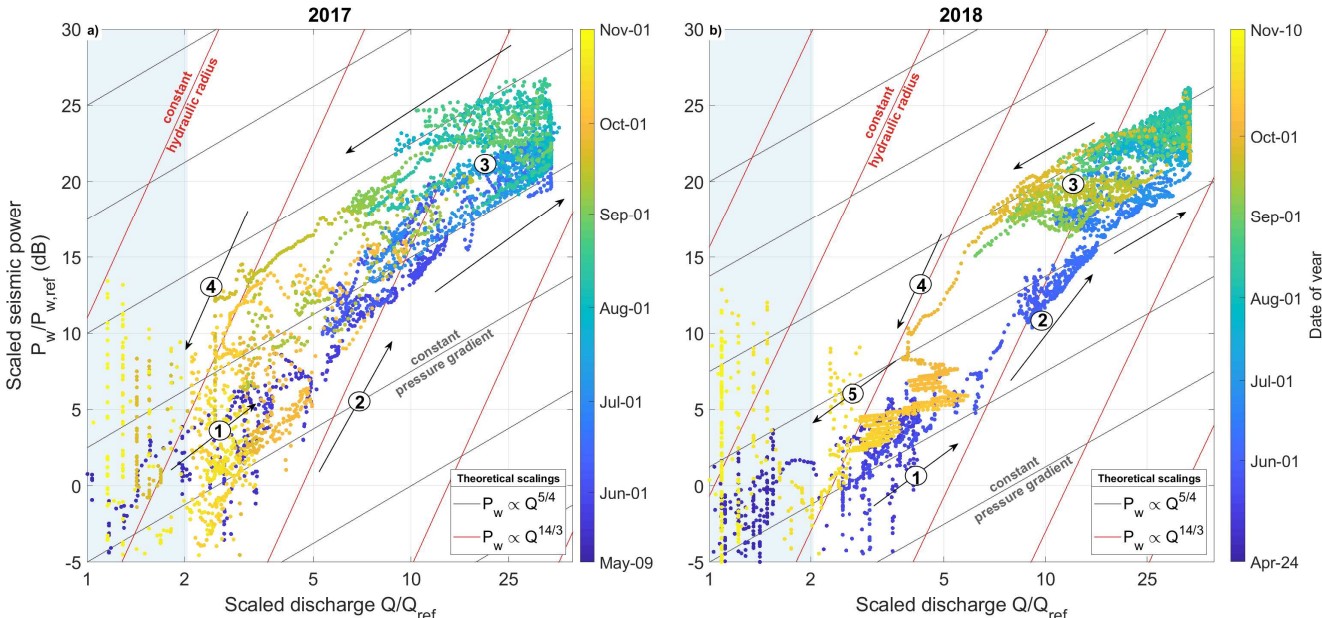

**Figure 6.** Observed and predicted changes in subglacial channel-flow-induced seismic power $\frac{P_w}{P_{w,ref}}$ versus changes in water discharge $\frac{Q}{Q_{ref}}$ during the melt season of years 2017 (a) and 2018 (b). Temporal signals are filtered with 1 h-lowpass filter. The color scale differs for the two years and varies with time from early April to Mid-November. Lines show predictions calculated from Eqs.(8) and (9) for constant hydraulic radii and varying hydraulic pressure gradient (red lines) and for constant hydraulic pressure gradient and varying hydraulic radii (black lines). Shaded blue areas represent the period when $Q$ is lower than $1\,\mathrm{m^3.s^{-1}}$. Arrows show the direction of time and circled numbers refer to periods described in the main text.

### 5.3.2 Analysis of diurnal changes

Observations and predictions of the diurnal relationship between the subglacial channel-flow-induced seismic power $P_w$ and water discharge $Q$ throughout the melt season are shown in Fig. 7. We quantify the diurnal behaviors over the two melt seasons by calculating the hysteresis amplitude $\phi$ and time lag $\delta t$ (see Sect. 4.3) and through comparisons of our observations with the theoretical predictions calculated for four selected days (panels (a) to (h) in Fig. 7). We selected these days for three reasons: they represent typical variations of $P_w$ and $Q$ over their respective periods ($\sim \pm 5$ days around their date); they show that our

observations capture diurnal variations from unique days without multi-days averaging; they give a pedagogical support for the reader to interpret values of the hysteresis amplitude $\phi$ and time lag $\delta t$.

The seasonal evolution of the diurnal hysteresis amplitude $\phi$ presents two peaks in late-May / early-June and in late-August / early-September, which are consistently observed in both 2017 and 2018 (phases ① in Fig. 7(i)). The seasonal evolution of the

diurnal time lag between $\delta t$ of $P_w$ to $Q$ presents is similar to that of $\phi$, with peak values at $\delta t > 2.5$ h in late-May / early-June



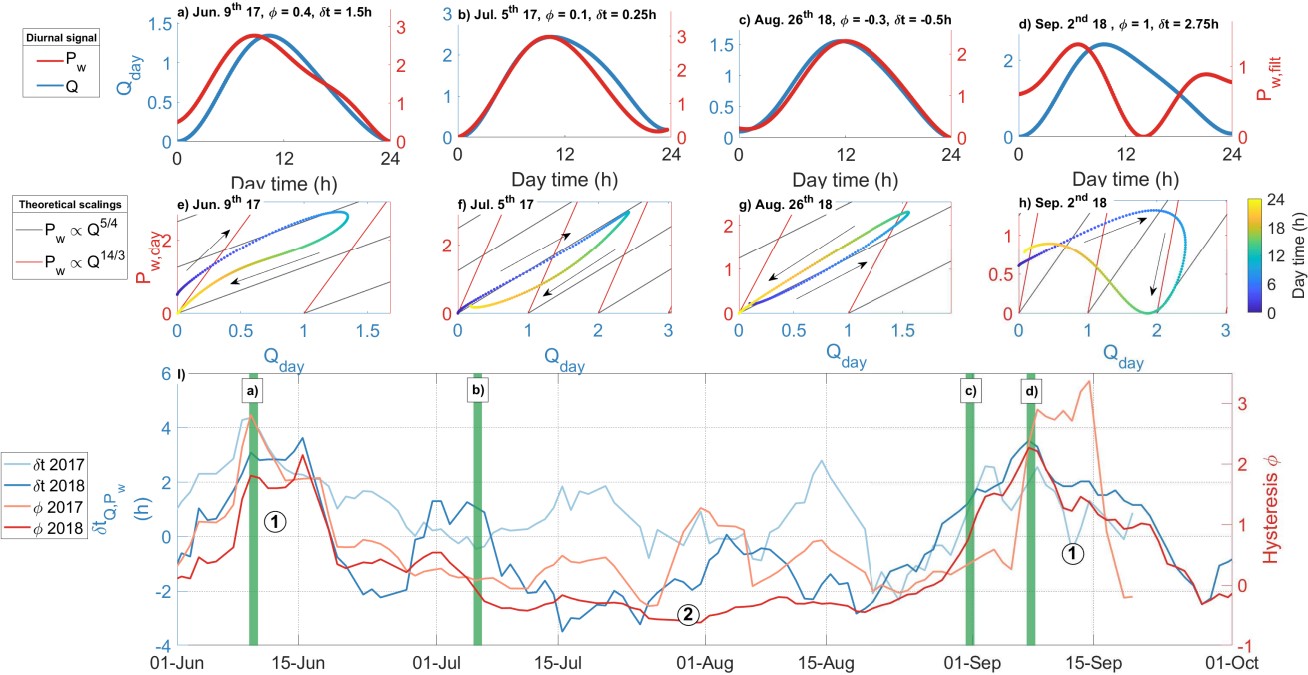

**Figure 7.** Diurnal observations of the subglacial channel-flow-induced seismic power $P_w$ and water discharge $Q$ and comparison with predictions from Gimbert et al. (2016). (a) to (d) Daily evolution of the [6-36] h bandpass filtered $P_{w,day}$ (red line) and $Q_{day}$ (blue line) for four selected days. Daily minima are removed for each day, setting the lower values at 0. (e) to (h) Observed (colored dots) and predicted (red and black lines calculated with Eqs.(6) and (7)) $P_w$ versus $Q$ daily relationships. Arrows show the direction of time. Note that yaxis bounds differs from panel to panel. (i) Diurnal time lag $\delta t$ between $P_{w,day}$ and $Q_{day}$ peaks (blue lines) and diurnal hysteresis $\phi$ between $P_{w,day}$ and $Q_{day}$ (red lines). Shaded lines are data of year 2017, plain ones of year 2018. Timeseries are smoothed over 10 days. Green vertical bars show times of the four selected days with the corresponding panel number. Arrow show the main trends and circled numbers refer to the two phases described in the main text.

and in late-August / early-September (Fig. 7(i)). This supports that hysteresis is mainly caused by phase difference between $P_w$ and $Q$ rather than by asymmetrical $P_w$ changes from rising to falling $Q$ (Sect. 4.3). Because the variability of $\delta t$ over the season is much larger than the predicted 0.04 h instrumental time lag (see Sect. 4.3), its evolution thus represents real changes in the relationship between $P_w$ and $Q$.


In the early and late melt season (phases ① in Fig. 7(i)), $P_{w,day}$ peaks more than 2 h before $Q_{day}$ and does present an asymmetrical shape with a steeper rising than falling limb (e.g. panels (e) and (h) of Fig. 7). This results in both a long time delay $\delta t$ and high $\phi$ values (Fig. 7(i)) due to a large clockwise hysteresis in $P_{w,day}$ versus $Q_{day}$. For example, on June $9^{th}$ our observations follow the trend of evolving hydraulic pressure gradient in the morning and the one of changing hydraulic

radius in the afternoon and at night. On September $2^{nd}$ our observations follow the trend of changing hydraulic radius in the





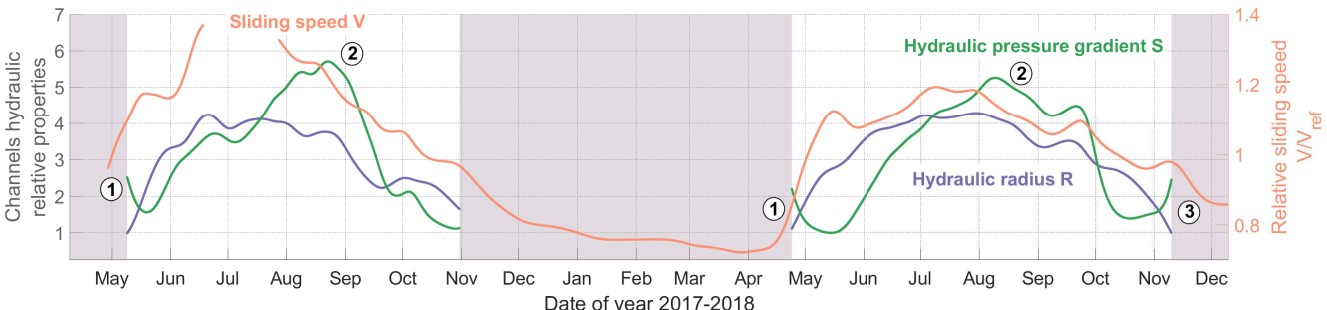

**Figure 8.** Seasonal evolution of the hydraulic radius $R$ and hydraulic pressure gradient $S$ as inverted from seismics as well as of glacier sliding speed as measured in-situ. (a) Relative hydraulic pressure gradient $\frac{S}{S_{ref}}$ (green line), relative hydraulic radius $\frac{R}{R_{ref}}$ (purple line) and relative sliding speed $\frac{V}{V_{ref}}$ (orange line). Shaded red areas represent the winter season. Temporal signals are filtered with 30 days-lowpass filter. Circled numbers described in the main text.

early morning and the one of evolving hydraulic pressure gradient in the afternoon. On the contrary, in summer (phase ② in Fig. 7(i)), both $\phi$ and $\delta t$ were at low minimum values of $\phi \simeq 0$ and $\delta t < -2$ h. At this time, $\delta t$ has a more pronounced seasonal and year-to-year variability than $\phi$ (Fig. 7(i)). In July and August (e.g. panels (b) and (c) in Fig. 7), $P_w$ peaks nearly at the same time as $Q$ with $\delta t < 0.5$ h and with an almost symmetrical diurnal evolution (Fig. 7(i)). For both summer days (July $5^{th}$ and

August $26^{th}$), our observations mainly follow the trend of changing hydraulic radius throughout the whole day, with a non-null hysteresis that shows that hydraulic pressure gradient may also change. This two-phases seasonal evolution shows that the early and late melt season diurnal changes in $Q$ cause a pronounced diurnal variability in the hydraulic pressure gradient and limited diurnal changes in the hydraulic radius, whereas over the summer channels show a more marked response to diurnal changes in $Q$ through changes in hydraulic radius.

**5.4    Inversions of changes in hydraulic radius and hydraulic pressure gradient**

We invert the relative changes of hydraulic radius $\frac{R}{R_{ref}}$ and hydraulic pressure gradient $\frac{S}{S_{ref}}$ using Eqs.(10) and (11) and our observations of $Q$ and $P_w$. In the following we use the notation $R$ and $S$ to refer to $\frac{R}{R_{ref}}$ and $\frac{S}{S_{ref}}$ for the sake of readability.

**5.4.1    Analysis of seasonal changes**

Observations of the temporal evolution of $R$, $S$ and the basal sliding speeds $V$ are presented in Fig. 8 to describe the seasonal

changes in both the channels and the cavities properties. All three variables show a well-marked seasonal evolution, with low values during the early and late melt season and high values in summer. However, differences between $R$, $S$ and $V$ exist over the melt season. For both years, $R$ starts increasing from the onset of the early melt season, until reaching a maximum within two months in Late-June to Early-July. $R$ is then four times larger than in the early melt season. In contrast, $S$ rapidly decreases in the first weeks of the melt season (Fig. 8 ①), concomitantly with an abrupt increase in $V$ by a factor of 1.5 compared to

winter. This shows that as the average water pressure rises in cavities and enhance sliding, channels on the contrary undergo





depressurization. The increase in $S$ then occurs with a delay of about one month compared to that in $R$, and $S$ reaches a maximum in August (Fig. 8 ②). $S$ is at that time five to six times larger than in mid-May. As $S$ increases, $V$ and $R$ have already attained their summer plateau. Contrary to the observations made on the Mendenhall Glacier (Alaska) where $S$ showed no significant trend over the two-month long investigating period (Gimbert et al., 2016), seasonal changes in water discharge at

the Glacier d'Argentière cause changes in both $R$ and in $S$. From early to mid-September, $R$ and $S$ decrease similarly and reach their minimum in late October. The summer to winter transition is most pronounced for $S$, which decreases by about a factor of 4 within less than a month (September to October) while $R$ decreases more gently. In 2018 for which we have an exploitable signal up to mid-November, we observe that $S$ increases again before winter, reaching values similar to those observed at the beginning of the melt season (Fig. 8 ③).


### 5.4.2 Analysis of diurnal changes

Figure 9 describes how channel and cavity properties behave at diurnal scale throughout the melt season. We quantify the diurnal behavior throughout the two melt seasons with the time lag $\delta t$ between $R$ and $Q$ daily maxima, noted $\delta t_{Q,R}$, and between $S$ and $Q$ daily maxima, noted $\delta t_{Q,S}$. We also calculate the amplitude of the diurnal variations $C_v$ for $R$, $S$ and $V$ (see Sect. 4.3

for definitions). In the same scopes as in Sect. 5.3.2 we illustrate in panels (a) to (d) in Fig. 9 the diurnal evolution of $R$ and $S$ for the same four selected days as in Fig. 7.

  The seasonal evolution of the amplitude of the diurnal variations of both $R$ ($C_v(R)$) and $S$ ($C_v(S)$) are similar and range from 5% at the season initiation to 15% in summer (Fig. 9(f)). In contrast, the seasonal evolution of $\delta t_{Q,R}$ and $\delta t_{Q,S}$ drastically differ

(Fig. 9(e)). On one hand, the temporal evolution of $\delta t_{Q,R}$ presents no marked changes throughout the season and remains within a range of $\pm$ 1 h (Fig. 9(e)) as highlighted by the four selected days (Figs. 9(a) to (c)). This shows that $R$ and $Q$ are consistently in phase on a diurnal basis throughout the melt season. On the other hand, the temporal evolution of $\delta t_{Q,S}$ presents two peaks of $\delta t_{Q,S} > 7$ h in June and August (Fig. 9(e) ①) and a period of low values ranging within [0;2] h in mid-summer (Fig. 9(e) ②). These changes in $S$ are clearly captured by the diurnal snapshots (e.g. Figs. 9(a) to (d)) that show a marked increase in

hydraulic pressure gradient in the morning before the rise in hydraulic radius. Such a difference in diurnal dynamics between $R$ and $S$ shows that channels exhibit high hydraulic pressure gradients in the early morning time while their hydraulic radius grows slowly to reach its maximum at the same time as the water discharge does.

  We also compare in Fig. 9(f) the diurnal dynamics of channel properties to the diurnal dynamics of the average water pressure conditions in cavities by comparing $C_v(R)$ and $C_v(S)$ with $C_v(V)$. Over the melt season, $C_v(V)$ exhibits a pattern that

is similar to $C_v(R)$ and $C_v(S)$, with higher values observed for the three variables in summer ($> 5$ %) than during the early and late melt season ($< 5$ %). This shows that short-term variability in channels properties (i.e. $R$ and $S$) correlates well with the short-term variability average water pressure condition in cavities. From late August to mid-September 2017, we observe that $C_v(S)$ reaches up to 15 % over less than a week, followed c. a week later by a rapid rise in $C_v(V)$ (Fig. 9(f) ③).

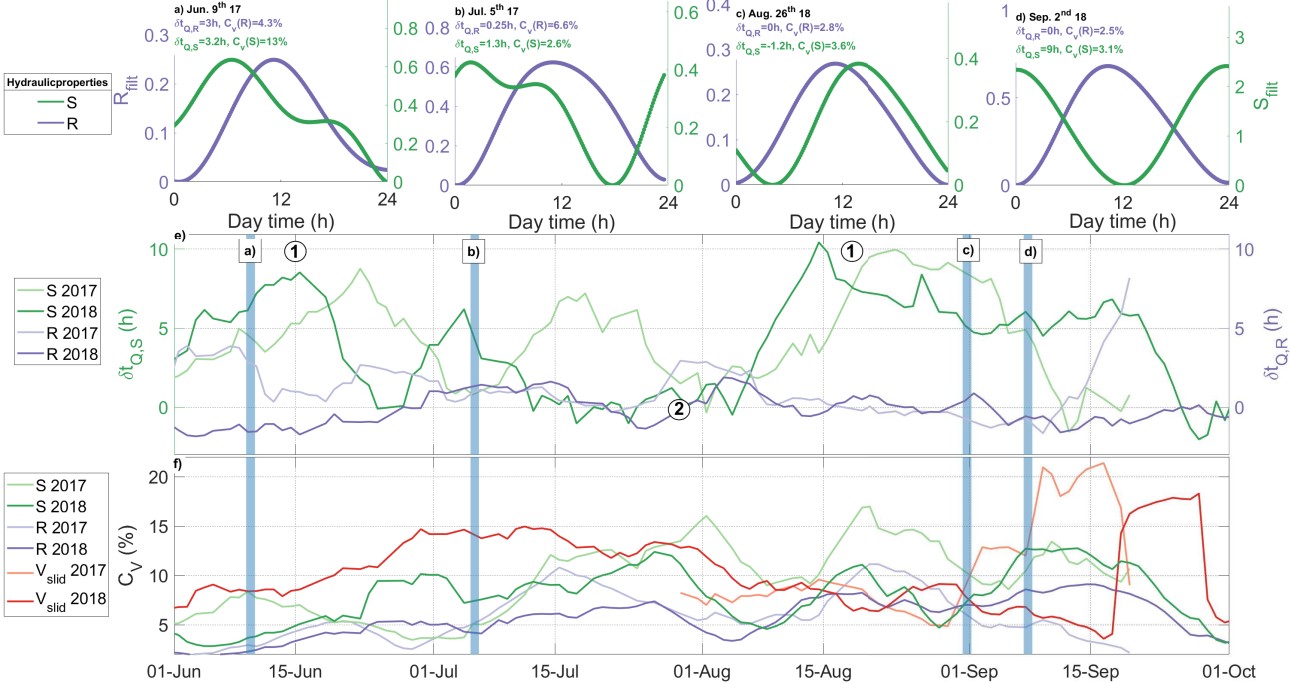

**Figure 9.** Diurnal evolution of the hydraulic radius $R$ and hydraulic pressure gradient $S$ and comparison to glacier dynamics. (a) to (d) Daily timeseries of $R$ (purple line) and $S$ (green line) for four selected days across the melt season. Timeseries are band-pass filtered within [6-36] h. Daily minima are removed for each day, setting the lower values at 0. Note that yaxis bounds differs from panel to panel. (e) Diurnal time lags $\delta t_{Q,R}$ between $R_{w,day}$ and $Q_{day}$ peaks (purple lines) and $\delta t_{Q,S}$ between $S_{w,day}$ and $Q_{day}$ peaks (green lines). (f) Sub-diurnal variability $C_v$ of $R$ (purple lines), $S$ (green lines) and the basal sliding speed $V$ (red line). Blue vertical bars shows location of the four selected days with the corresponding figure number. Shaded lines are data of year 2017, plain ones' data of year 2018. Circled numbers described in the main text.

## 5.5 Comparison of inversions with predictions from Röthlisberger (1972)

Our seismically inferred $S$ and $R$ values are shown in Fig. 10 as a function of water discharge $Q$, along with scaling predictions calculated using the theory of Röthlisberger (1972) assuming channels at equilibrium (melt rate equals creep rate) with $S \propto Q^{-2/11}$ and $R \propto Q^{9/22}$ (Eqs.(14) and (12), green lines in Fig. 10) and channels out-of-equilibrium that respond to changes in $Q$ only through changes in $S$ with $S \propto Q^2$ and $R$ constant (Eq.(13), purple lines in Fig. 10). Our observations present two distinct regimes. At low discharge during the early and late melt season (Fig. 10 ①) our observed changes in $S$ and $R$ with

$Q$ are well predicted by theory for channels behaving at equilibrium. At high discharge (mid-May to early October, Fig. 10 ②) changes in $S$ and $R$ with $Q$ significantly departs from predictions of channels at equilibrium and approaches the one of channels evolving out-of-equilibrium through changes in $S$ solely. The transition between the two regimes herein observed is quite abrupt for $S$ which rapidly switches from being a decreasing to being an increasing function of $Q$. For $R$, the transition is





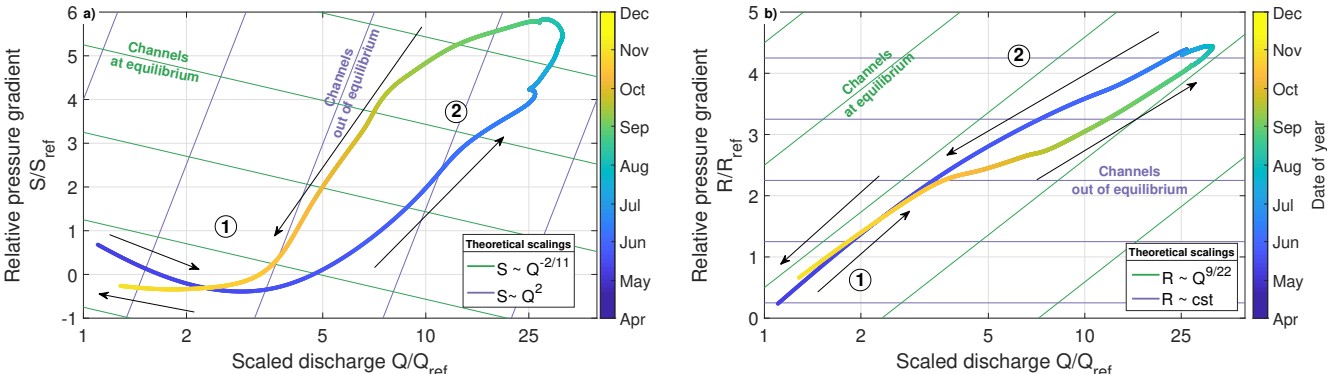

**Figure 10.** Inverted (a) relative hydraulic pressure gradient $\frac{S}{S_{ref}}$ and (b) hydraulic radius $\frac{R}{R_{ref}}$ shown with respect to the relative water discharge $\frac{Q}{Q_{ref}}$. All variables are first averaged over the 2017 and 2018 melt season then 60 days-period highpass filtered and then taken relative to their respective value at the end of the melt season. The observations are compared to the prediction of Röthlisberger (1972) in the case of subglacial channel evolving at equilibrium with $Q$ (green lines, $S \propto Q^{-2/11}$ and $R \propto Q^{9/22}$) and for subglacial channel evolution dominated by hydraulic pressure gradient changes (blue lines, $S \propto Q^2$ and $\frac{\delta R}{\delta Q} = 0$). Arrows show the direction of time and circled numbers refer to periods described in the main text.

marked by a weaker dependency on $Q$. During the period when $Q/Q_{ref} > 5$ best datafit of $R$ to $Q$ gives $R \propto Q^{11/44}$ and for the

periods when $Q/Q_{ref} < 4$ it gives $R \propto Q^{9/22}$. This latter scaling is in close agreement with the predicted scaling of $R \propto Q^{9/22}$ calculated using the theory of Röthlisberger (1972) assuming channels at equilibrium.

## 6    Discussion

### 6.1    Evaluating bias from potential changes in the number and position(s) of channel(s)

As stated in Sect. 2, subglacial channel-flow-induced seismic power $P_w$ depends on the number of subglacial channels $N$

(Eqs.(10) and (11)) and on the source-to-station distance, which we both considered as constant in this study. Here we discuss how much potential changes in channel(s) positions and in $N$ may bias our inversions of $S$ and $R$. On one hand, given the glacier configuration in our study area (250 m thick, 500 m wide Fig. 2(a)), channels-to-seismic station distance is similar regardless of whether channels are located at the glacier center or on its sides. Therefore, we do not expect changes in channel spatial positions to bias our inverted values of $R$ and $S$. On the other hand, we estimate how much observed changes in $P_w$

would require changes in $N$ if they were to be explained only by an evolving number of channels rather than evolving $S$ or $R$. From Eq.(10) we have that $S$ weakly depends on $N$ compared to on $P_w$ and on water discharge $Q$. As a result, explaining $P_w$ variations while imposing $S$ constant would require $N$ to change by more than 4 orders of magnitude ($5^{41/6}$), which is unrealistic. From Eq.(11) we have that $R$ weakly depends on $N$ compared to on $Q$. As a result, explaining $P_w$ variations while imposing $R$ as constant would require $N$ to change by more than factor of 30 ($4^{-82/33}$), which is also likely unrealistic since





at the onset of the melt season channels are expected to form an arterial network with few channels being kept over summer (Schoof, 2010; Werder et al., 2013). Therefore, we do not expect potential changes neither in channel positions nor in $N$ to cause significant bias in our inverted values of $R$ and $S$.

## 6.2   Implications for inferring water discharge using seismic noise

As opposed to Gimbert et al. (2016) who observed little variations in hydraulic pressure gradient over its two-month long period of observations on the Mendenhall Glacier, on the Glacier d'Argentière we observe high and sustained channel pressurization over the whole summer (June-September). This has implications for the physics of subglacial channels, which we further discuss in Sect. 6.3, and also for our capacity to invert for discharge $Q$ based on observed seismic power $P$. If one considers the equilibrium assumption over the melt season this yields, under Röthlisberger (1972) steady-state equilibrium assumptions,

to the scaling $Q \propto P_w^{33/31}$ (see Eqs.(6) and (12)). When applied over the melt season using our observations of $P_w$ at Glacier d'Argentière, this underestimates the measured discharge by more than 65%. As shown in Fig. 10, such assumption is only valid for the early and late melt season when both discharge and its variability are low. Using the approximation $Q \propto P_w^{33/31}$ may be more appropriate for periods of low discharge and settings with limited water supply variability such as in Antarctica. If one now considers the empirical relationship $Q \propto P_w^{18/43}$ obtained from the period of channels being out of equilibrium (using

Eq.(6) and $R \propto Q^{11/44}$, see Sect. 5.5), this leads to an uncertainty of less than 10% on the estimated water discharge over the melt season at Glacier d'Argentière. We therefore suggest that the $Q \propto P_w^{18/43}$ relationship may be preferred for periods of high discharge and settings with strong seasonal variability in water supply (e.g. Alpine and Greenland glaciers). For such settings, our relationship could therefore allow to invert for the water discharge simply using on-ice seismic instrumentation rather than direct measurements of the water discharge.

## 6.3   Implications for subglacial hydrology and ice dynamics

### 6.3.1   Using periods when channels are in equilibrium to estimate channel(s) size and number

During the early melt season when $Q$ is on the order of 1 m$^3$.s$^{-1}$ (Fig. 5), channels are observed to be at equilibrium with changes in $Q$ (Fig. 10). This behavior supports that the channel's hydraulic capacity is sufficient to accommodate water input at this time of the year. We propose that, at those times, changes in water supply occur at a rate that is lower than that at which

channels adjust their hydraulic radius. During the early melt season, low rates in water input changes are likely caused by water supply from melt being highly damped by the snow cover (Marshall et al., 1994; Fleming and Clarke, 2005). During the late melt season, the cause of low rates in water input is less clear. We suggest that such rates could be induced by englacial stored water being slowly released (Flowers and Clarke, 2002; Jansson et al., 2003). Because of the well developed drainage system at those times, channels could also adjust faster their hydraulic radius than during the early melt season and therefore could

behave at equilibrium for higher rates in water input than during the early melt season.





Using Eqs.(6) and (8) of Hooke (1984) that predict the conditions of equilibrium for steady-state channels and assuming that total discharge is equally distributed over channels of identical geometry (R-channels), we find that in our case equilibrium is predicted if the number of channels lies between 4 and 6 (using an ice thickness of 250 m, a down-glacier surface slope of

$5°$ and a total water discharge of $1 \text{ m}^3.\text{s}^{-1}$; see Appendix Sect. B). For a lower (resp. higher) number of channels, discharge per channel and thus channel-wall melt is higher (resp. lower) than the expected channel-wall creep, which violates the equilibrium condition. Our estimate of 4 to 6 channels is consistent with the numerical modelling results of Werder et al. (2013) of 4 to 5 dominant channels lying below the Gornerglestcher tongue (CH), a glacier which has a similar geometry to that of the tongue of Glacier d'Argentière (c. 500 m wide, c. 300 m maximum thickness). Further insights on the spatial evolution of

the subglacial drainage system could be gained using seismic arrays to locate the source(s) of subglacial flow-induced-seismic noise (Lindner et al., 2019).

We propose to estimate the size channel at the season initiation based on the channel number previously proposed. With $5 \pm 1$ channels and $1 \text{ m}^3.\text{s}^{-1}$ equally distributed discharge, the average discharge per channel is of about $0.20 \pm 0.05 \text{ m}^3.\text{s}^{-1}$

(uncertainty is obtained from that on channels number). Considering that subglacial flow-induced-seismic noise is sensitive to water flow speeds on the order of $1 \text{ m.s}^{-1}$ (Gimbert et al., 2016) we can estimate a minimal channel cross-section area of about $0.20 \pm 0.05 \text{ m}^2$, and a resulting channel radius of $0.35 \pm 0.05 \text{ m}$ (for semi-circular R-shaped channels). We note that absolute inversions of $R$ and $S$ could be done by explicitly formulating the Green function $G$ in Eq.(1), and compared to the present estimation using channels at equilibrium. However, this is beyond the scope of this study.

**6.3.2    Understanding highly pressurized channels during the plain melt season**

At discharges higher than $1 \text{ m}^3.\text{s}^{-1}$ the hydraulic pressure gradient $S$ in channels remains high (Fig. 10). Considering that bed slope is constant, these high $S$-values require channels to be full and pressurized. During these periods of high discharge, as $S$ increases with the water discharge $Q$ (Fig. 10(a)) channels respond to changes in discharge in a comparable way as the cavities theoretically described by Schoof (2010). Such an observation is opposed to the observations of Andrews et al. (2014) made

in Greenland and to the theoretical steady-state predictions of Schoof (2010) and Werder et al. (2013) that instead suggest channels to have a decreasing water pressure as channels develop over the summer.

Using Hooke (1984) and our estimate of 5 channels made in Sect. 6.3.1, we find that in our case channel-wall melt (i.e. opening rate) is expected to dominate ice creep (i.e. closing rate) for $Q > 1 \text{ m}^3.\text{s}^{-1}$ (see Sect. B for details on the calculation).

At steady-state this should either lead to channel growth and/or $S$ abrupt decrease if free-flow (i.e. atmospheric pressure) is reached. These two scenarii are not observed during summer since $R$ stays mainly constant (i.e. limited channel growth) and $S$ presents high values supporting closed-flow over hourly timescales. We propose that the summer channel pressurization (high $S$) is linked to the channel's response to the marked diurnal and short-term variability in the water supply (as theoretically described in Schoof (2010)), and that channels behave out-of-equilibrium because changes in water input occur at a rate that is





much higher than that at which channels can adjust their hydraulic radius.

This interpretation is supported by diurnal observations. In the morning, $S$ is observed to rise earlier than $R$ (Fig.9), suggest-ing that channel-wall melt does not accommodate the increase of $Q$ fast enough and causes pressurized flow. As water supply increases, channels start to respond to the water input and grow by channel-wall melt leading to a delayed hydraulic radius $R$
increases compared to $S$ (Fig. 9). At the same time the channel capacity increases with $R$ (Röthlisberger, 1972) leading to a decrease in $S$ before the $Q$ peak as observed in Fig. 9. During the afternoon, as the water supply decreases, $R$ slowly decreases by much less than a percent per hour (Fig. 9). At this rate, ice creep is capable to adjust $R$ fast enough to limit open channel-flow (Fig. S5). This is consistent with our observation that does not show an abrupt decreases in $S$ as one expects if open channel-flow occurs (Fig. 9). The hydraulic pressure gradient therefore builds up from day-to-day over the summer. During
night-time, as $Q$ is at its minimum, the closure rate still adjusts channel size and therefore allows $R$ to remain nearly constant through summer. This proposed scenario is consistent with both the observed diurnal dynamics in the hydraulic properties and may explain the unexpected pressurized channels during summer. Estimation of melt and creep rates calculated from Hooke (1984) in a similar manner as in Sect. 6.3.1 supports the plausibility of such diurnal dynamic (see Appendix Sect. B for details).

### 6.3.3   Channel dynamics, cavity water pressure and basal sliding

Our observations (Figs. 8 and  10) indicate that over the summer channels are pressurized and behave out-of-equilibrium with the water input. On the other hand, during summer the glacier sliding speeds remain high, especially in 2018, (Fig. 5), which shows that the average basal water pressure (that is mainly set by pressure in the cavities) is also high. These concomitantly high pressures in channels and in cavities suggest that the two systems may be connected.


During summer, because of channel-flow pressurization, the channel-system does not operate under a significantly lower hy-draulic potential than that of the cavity-system. This would therefore prevent significant water flow from cavities to channels, and leads to cavities that are kept pressurized. This sustained high water pressure at the glacier basis favors high glacier sliding speeds over summer. Such channel-cavity-sliding link, has been previously suggested (Hubbard and Nienow, 1997; Andrews
et al., 2014; Rada and Schoof, 2018) but was not based on independent observations of cavities and channels as done presently.

We suggest that during these periods of pronounced short-term variability in water supply, the whole drainage system be-comes well-connected although with a limited drainage capacity. Thus the channel system may participate in maintaining high pressure in cavities and thus high sliding speeds during periods of high water supply variability. Short-term variability in water
supply may lead to pronounced glacier acceleration even during situation of a well-developed channel network. Such sub-glacial hydrology/ice dynamics link deserves further investigation through combination of seismic observations and subglacial hydrology/ice dynamics models (e.g.  Gagliardini and Werder, 2018). Indeed a better understanding of the impact of short-lived water input on glacier dynamics is necessary as under climate warming short-term climatic variability and extreme event





occurrences are expected to increase (Hynčica and Huth, 2019), potentially causing greater glacier acceleration than previously
thought (e.g. Tedstone et al., 2015).

## 7    Conclusions

We investigate the physics of subglacial channels and its link with basal sliding beneath an Alpine glacier (the Glacier
d'Argentière, French Alps) through the analysis of a unique two-year long dataset made of on-ice measured subglacial water-
flow-induced seismic power and in-situ measured glacier basal sliding speed records. Our study shows that the theory of Gim-
bert et al. (2016) is consistent with our observations and that the analysis of the seismic power measured within the [3-7] Hz
frequency range allows to study the subglacial drainage properties over a complete melt season and down to diurnal timescales.

We quantify temporal changes in channels hydraulic radius and hydraulic pressure gradient using the theory of Gimbert
et al. (2016) and measurements of water discharge concomitant to our seismic record. Our approach allows to isolate sub-
glacial water-flow-induced seismic power from that of other seismic sources, and makes possible observing changes at various
timescales (from seasonal to hourly) and water discharge ranges (from 0.25 to 10 $\mathrm{m}^3.\mathrm{sec}^{-1}$). At seasonal timescales we ob-
serve, for the first time, that hydraulic radius and hydraulic pressure gradient both present more than a four-fold increase from
spring to summer, followed by a comparable decrease towards autumn. Comparing our observations to the theoretical predic-
tions of Röthlisberger (1972) we identify that channel dynamics over the season is characterized by two distinct regimes yet
unprecedentedly reported. At low discharge during the early and late melt season we observe that channels respond to changes
in discharge mainly through changes in hydraulic radius, and that the strong changes in hydraulic radius and weak changes in
pressure gradient are well predicted by theory for channels behaving at equilibrium. We propose that, at those times, changes
in water input occur at a rate that is lower than that at which channels adjust their hydraulic radius. During the early melt
season, these low rates in water input changes are likely caused by water supply from melt being highly damped by the snow
cover. From this equilibrium channel-dynamics condition we are able to estimate the number of channels, which we find to be
between 4 to 6, each channel having a radius of about 0.5 m in the early melt season that may go up to 2 m in summer. At high
discharge and high short-term water-supply variability (often during summertime) we observe that channels undergo strong
changes in hydraulic pressure gradient, a behavior that is not expected for channels at equilibrium. Instead, those changes
in hydraulic pressure gradient are well reproduced by theory under the end-member consideration of no changes in channel
geometry in response to changes in water input. We propose that, at those times, channels behave out-of-equilibrium because
changes in water input occur at a rate that is much higher than that at which channels adjust their hydraulic radius. This in-
terpretation is supported by observations at the diurnal scale, which show that channels pressurize in the early morning and
depressurize in the afternoon as their hydraulic radius slowly grow concomitantly with the water supply rise. At night when
water discharge decreases, ice creep then allows channels to recover their initial early morning hydraulic radius. We do not
observe significant decrease of the hydraulic pressure gradient during those days, which indicates that the hydraulic pressure
gradient builds up from day-to-day concomitantly to a hydraulic radius that is kept nearly constant. Channels may thus remain





pressurized over the whole summer because of the short-term (diurnal, rain) variability in water supply, which forces channels to respond through a transient-dynamic state.

Channels behaving out-of-equilibrium during most of the melt season also has implication for the use of subglacial water-flow-induced seismic power $P_w$ to invert for water discharge $Q$. The empirical relationship between $Q$ and $P_w$ that we derive during the period when channels are out-of-equilibrium allows estimating a water discharge from seismic noise with an error of less than 10 %, while an error of 65 % is obtained when assuming channels at equilibrium. Our presently proposed out-of-equilibrium relationship for inverting discharge could be applied in settings with strong seasonal variability in water supply

(e.g. Alpine and Greenland glaciers). During summer we also observe high and sustained basal sliding speeds, supporting that the widespread inefficient drainage system (cavities) is likely pressurized. We propose that channels being also pressurized may help sustain high pressure in cavities and thus high glacier sliding speeds.

    These results demonstrate that on-ice passive seismology is an efficient tool to overcome the classical observational limita-

tions faced when investigating subglacial hydrology processes. In this respect, our results bring new constraints on channels physics, on links between channels, cavities and sliding, and on the use of passive seismology to invert for subglacial water discharge. Using the two regimes herein observed in channels seasonal-dynamics as constraints for subglacial hydrology/ice dynamics models may allow to strengthen our knowledge on the physics of subglacial processes. We therefore encourage the subglacial hydrology/ice dynamics modeling community to consider these newly-seismically-derived observations.

*Code and data availability.*    The presented dataset will be made publicly available in the future. Ongoing work is taking place to meet the format and documentation required for the release, which is expected to happen fully or partially by mid-2021. In the meantime, it is available on request from the corresponding author. The Python and SAC codes for seismic power calculation are given in the Supplementary Materials.

## Appendix A:  Frequency content of the water discharge and the subglacial channel-flow-induced seismic power

We show in Fig. A1 the power spectrum of the water discharge $Q$ (blue lines) and subglacial channel-flow-induced seismic

power $P_w$ as a function of the period. We observe for both variables a well-defined peak at one day and 12 h period. This shows that these signals present a clear diurnal and sub-diurnal variability, and supports our choice to band-pass-filter these signals within [6-36] h to study these short-term variabilities.

## Appendix B:  Evaluating theoretical melt and creep rates with Hooke (1984)' equations

We used in this study the equations 6 and 8 of Hooke (1984) to evaluate the theoretical melt rate $\dot{m}$ and creep rate $\dot{r}$, as follows



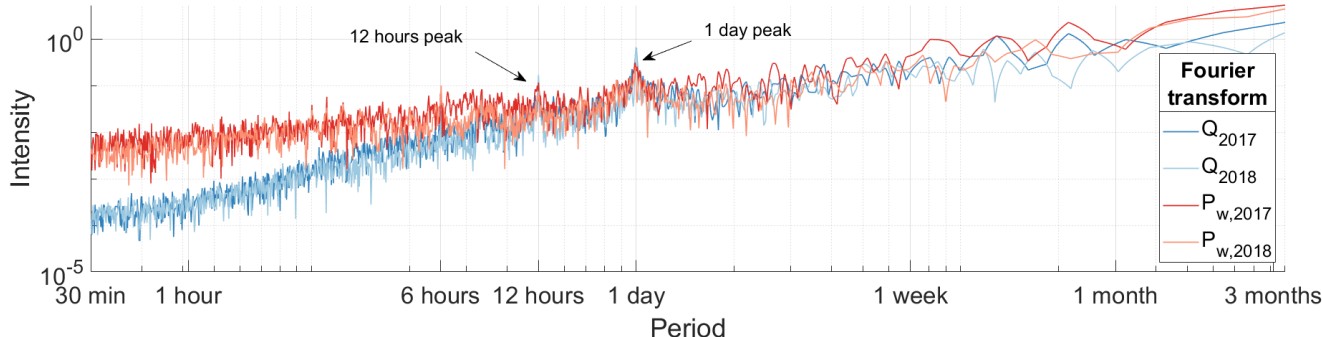

**Figure A1.** Power spectrum of the water discharge $Q$ (blue lines) and subglacial channel-flow-induced seismic power $P_w$ (red lines) shown a function of the period. Both axis are in logarithmic scale (1 over the frequency.)

$$\dot{m} = C_2 Q^{3/5} sin(\beta)^{6/5}, \tag{B1}$$

$$\dot{r} = C_3 \frac{Q^{2/5}}{sin(\beta)^{1/5}} H^3, \tag{B2}$$

with $H$ the ice thickness, $\beta$ the down-glacier surface slope, $C_2$ and $C_3$ constant. We use the values of Hooke (1984) for the two constants: $C_2 = 3.731e^{-5}$ m$^{-4/5}$ s$^{-2/3}$ and $C_2 = 5.71e^{-14}$ m$^{-16/5}$ s$^{-3/5}$. For the glacier geometry we use using an ice thickness of 250 m and a down-glacier surface slope of 5°.

*Author contributions.* U.N., F.G. and C.V. designed the study. U.N. performed the seismic analysis with input from F.G. and F.W.. U.N. interpreted the results with input from F.G.. U.N. led the writing of the paper and F.G., C.V., F. W. and D. G. contributed to it. L.P and L.M were in charge of the basal sliding speeds measurements. All authors participated to field installations.

*Competing interests.* The authors declare that they have no competing interests.

*Acknowledgements.* We thank J. Bolibar, C. Bouchayer, J. Brives, J. Brondex, J. Chowdhry, S. Escalle, A. Gilbert, B. Jourdain, O. Laarman, B. Lipovsky, N. Maier, A. Palenstijn, O. Passalacqua, L. Preiswerk, A. Rabatel, V. Ramseyer, V. Tsai, B. Urruty and J. Wille for assistance in the field. We are indebted to Electricité Emosson SA (hydroelectric company) for the water discharge measurements and access to the subglacial galleries and the French GLACIOCLIM project for temperature and precipitation data (https://glacioclim.osug.fr/). We thank Nikos Germenis for technical support on the ARG seismic network (https://geobit-instruments.com/). U. Nanni would like to thank J. Bolibar, B. de Fleurian and L. Preiswerk for fruitful discussions. This work was supported by the SEISMORIV project (ANR-17-CE01-0008) and





the SAUSSURE project (ANR-18-CE01-0015) funded by the Agence National de la Recherche (ANR). U. Nanni is funded by the French
Ministère de l'Enseignement Supérieur, de la Recherche et de l'Innovation (MESRI).



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
