# Peer review of "Quantification of seasonal and diurnal dynamics of subglacial channels using seismic observations on an Alpine Glacier."

_The Cryosphere, 2019_

## Referee Comment (RC1) · Anonymous Referee #1 · 21 Dec 2019

This manuscript presents a two-year long dataset of seismic measurements together with sliding velocity records made on a glacier in the French Alps, with the aim of characterising the dynamics of subglacial channels over seasonal and diurnal timescales. In my opinion, it is overall well and clearly written, although I found some of the notations confusing in places; the data interpretation is sound and leads to very interesting results. In particular the authors conclude that while channels behave at equilibrium when the discharge is low, they switch to an out-of-equilibrium regime at high discharge, enabling fast ice velocities throughout the summer (e.g., channels do not necessarily have the ability to regulate the ice flow, as anticipated).

I support the publication of the manuscript with minor corrections, and my comments below are aimed at improving clarity further.

[Figure]

-L.53: Interactions between channels and cavities have never been observed. I suggest replacing "observed indirectly" with "inferred".

-L. 55: similarly, suggest replacing "observed to trigger " with "linked to"

-L.75: remove parenthesis around Gimbert et al.

-L.121: Should the Manning-Strickler relation read V= . ... (rather than U=. . .)?

-L. 140: It would be helpful to have more details on how the authors went from Eqs. 6-7 to Eqs. 10-11, which as stated, are difficult to follow. I suggest adding details in the supplementary material, or as appendix.

-L.174: Suggest replacing "water discharge routing subglacially" with "Subglacial water discharge" ?

-L. 177: By assuming that discharge at the snout is representative of the discharge routed under the seismometers, you are basically assuming that upstream water flow largely coincide with your seismometers locations. This might be correct, but I don't quite see how you can be certain of the location of the upstream subglacial path. For example, have you tried to route the water according to hydraulic potentials?

-L.230-235: A bit hard to follow. Why is Pa set to 0 dB here? Isn't Pmea-PA just Pw, and if so, why not just using Pw? Perhaps what is confusing here, is that you have two ways of evaluating Pw; one from the discharge Q, as plotted on Figure 5; and one from subtracting an estimate value of PA from a measured Pmea. Unless I am just missing something, it could be clearer to use different notations, or explain better how each quantity is used in the study (for example, observed vs interpreted). Similarly, the first line in section 5.2 (l.297) refers to the red line shown on Figure 5b as Pmea, while the caption indicates that it is Pw. More consistency is needed throughout to increase the clarity of what has been done.

-L. 242: three appropriate metrics? (rather than appropriated?)

-L. 262: Is P the same as Pmea? If it is, make sure the notations are consistent throughout. If not, explain somewhere how they differ.

-L. 284: The definition of the summer period varies throughout the manuscript. Sometimes it is August to mid-october, sometimes in includes July (e.g. line 364), yet again different in line 385, or 447… this needs to be more consistent, if one wants to attribute processes to specific seasons. Or use individual months and refer to a specific period.

-L. 294: typo in "appear to not significantly" – (remove "do")

-L. 318-319: week day, week end, holidays ?? that was confusing.

-L. 350: delete "presents"

-L. 371: What are the values for Rref and Sref? (and Vref, Qref, mentioned elsewhere) – Also, I find the chosen notation of R for R/Ref and of S for S/Sref (etc…) confusing. As it stands, it is not always clear which variable is being referred to or analysed.

-L. 374: Is V supposed to be V/Vref (as used on Figure 8)?

-L. 409: As I understand, the average basal water pressure is used as representative of pressure conditions in cavities (c.f. L.524). This could be explained here, as it is otherwise unclear how you quantified the latter.

L.462: Again, more clarity needed wrt notations: here, there are two references to Q in the same sentence, one being from Figure 5 showing the measured water discharge, and the other being to the scaled discharge shown on Figure 10.

L.490: "the plain melt season" doesn't read well…

Figure 10: should the Y axis on panel B refer to hydraulic radius?

---

## Referee Comment (RC2) · Anonymous Referee #2 · 1 Jan 2020

The work by Nanni and co-workers represents a significant contribution to our understanding of using glacial seismology to monitor glacial hydrology. This work utilizes seismic data and observations of subglacial discharge acquired over two full melt season. They then use the theory of Gimbert et al. (2016) to investigate of seasonal variations in seismic tremor can be used to make inferences about seasonal to daily variation in the subglacial hydraulic system.

I find the significant results of the this paper to be: 1) Significant advances in documenting methodology for conducting this sort of analysis. For example, robustly combing multiple seismic records to forms continuous measure of seismic tremor, differentiating anthropogenic noise from glacial tremor (figure 3), and presenting quantitative measures of hysteresis (i.e., equations 15-17)

[Figure]

2) Demonstrating that seasonal variations exist between seismic power and discharge (Figure 6) which are most likely related to changes in the subglacial hydraulic system and that the relationship between seismic power and discharge varies depending on the season (section 6.2)

3) Demonstrating how the theory of Gimbert et al. (2016) can be used to investigate daily to seasonal variations in subglacial dynamics. However, occasionally the authors refer to their model derived parameters as observations when in fact they are a model result. For example on line 462, the authors use the phrase "...channels are observed...", the channels are in fact not observed but "...channels are inferred to be at equilibrium..." based on the theory of Gimbert et al. (2016). Other instances occur at line 551-555. To summarize, I think it is important to remain clear that the derived values of hydraulic radius (R) and pressure gradient (S) are in fact NOT observations but model derived parameters.

Other Comments:

Diurnal Variability, Hysteresis, and Phase lags: Figure 7 clearly shows the seasonal variability of the phase lag and hysteresis in Pw and Q. However, I was surprised to see no measure of coefficient of variation (Cv) as is done in figure 9. Figure S3 appears to show that Q often does not display diurnal variability. Is this true or is this just a figure resolution issue? If there are days when there is no diurnal variability in Q, what is the meaning of the phase lag or the hysteresis measurements?

Detail Comments: Title: Should include a mention of seismology.

Line 1: should be "...knowledge of the..."

Line 165: remove "days"

Line 173: perhaps "Subglacial water discharge is monitored.."

Line 186 (and elsewhere). I believe that Bar should be replaced with Pa.

[Figure]

Line 197:perhaps "borehole type sensors"

Line 203: should be "A few data gaps..."

Line 230: specify as synthetic.

Figure 5: in caption rephrase "Shaded red area" as "Red shaded area" and "Shaded blue area" as "blue shaded area"

Line 285-289: Specific days need to be specified.

Line 298: I would prefer that "in response" should be changed to something like "correlates with". The phrase in "in response" is an interpretation that already presumes Q is the driver of Pw.

Figure 6 ( and line 371): Over what time period is Qref and Pwref defined?

Line 371: "We invert for ...."

Figure S3: I think that the legend has Pa and Pw switched.

---

## Author Comment (AC1) · 11 Feb 2020

We thank reviewer 1 for his general comments and very useful suggestions.

You can find attached our detailed responses to the comments (NANNI_response_RC1_tc_2019_243.pdf) together with the changes made in the manuscript, with in red what has been removed and in blue what has been added (NANNI_2020_diff_revisions_tc_2019_243.pdf).

We also attach general comments concerning changes that were not made following a particular comment of RC1, but that answer questions that arose when working on the reviews (NANNI_response_general_tc_2019_243.pdf).

Best, Ugo Nanni in behalf of the co-authors

[Figure]

Please also note the supplement to this comment:
https://www.the-cryosphere-discuss.net/tc-2019-243/tc-2019-243-AC1-supplement.zip

––––––––––––––––––––––––––––––

---

## Author Comment (AC2) · 11 Feb 2020

We thank reviewer 2 for his general comments and very useful suggestions.

You can find attached our detailed responses to the comments (NANNI_response_RC2_tc_2019_243.pdf) together with the changes made in the manuscript, with in red what has been removed and in blue what has been added (NANNI_2020_diff_revisions_tc_2019_243.pdf).

We also attach general comments concerning changes that were not made following a particular comment of RC2, but that answer questions that arose when working on the reviews (NANNI_response_general_tc_2019_243.pdf).

Best,

[Figure]

Ugo Nanni in behalf of the co-authors

Please also note the supplement to this comment:
https://www.the-cryosphere-discuss.net/tc-2019-243/tc-2019-243-AC2-supplement.zip

---

## Author Response (AR1)

Note to the editor:

As no changes have been asked by the reviewers the manuscript (pdf) has not been changed since the version sent to the reviewers. We have only added a link in the data availability section to 10.5281/zenodo.3701520 where we made available a part of the data used in this study.

Supplement materials have not been changed since the version sent to the reviewers.

Thank you for your editing,

Sincerely

Ugo Nanni in the behalf of the co-authors

26/03/2020